# Application of the Luminescence Syncytium Induction Assay to Identify Chemical Compounds That Inhibit Bovine Leukemia Virus Replication

**DOI:** 10.3390/v15010004

**Published:** 2022-12-20

**Authors:** Hirotaka Sato, Jun-na Fukui, Hiroyuki Hirano, Hiroyuki Osada, Yutaka Arimura, Michiaki Masuda, Yoko Aida

**Affiliations:** 1Department of Microbiology, School of Medicine, Dokkyo Medical University, Tochigi 321-0293, Japan; 2Virus Infectious Diseases Unit, RIKEN, Saitama 351-0198, Japan; 3Department of Host Defense for Animals, School of Animal Science, Nippon Veterinary and Life Science University, Tokyo 180-8602, Japan; 4Chemical Resource Development Unit, RIKEN Center for Sustainable Resource Science, Saitama 351-0198, Japan; 5School of Pharmaceutical Sciences, University of Shizuoka, Shizuoka 422-8526, Japan; 6Laboratory of Global Infectious Diseases Control Science, Graduate School of Agricultural and Life Sciences, The University of Tokyo, Tokyo 113-8657, Japan

**Keywords:** bovine leukemia virus, LuSIA (luminescence syncytium induction assay), high-throughput screening, inhibitor, BSI-625, BSI-679, infection

## Abstract

Bovine leukemia virus (BLV) infection causes endemic bovine leukemia and lymphoma, resulting in lower carcass weight and reduced milk production by the infected cattle, leading to economic losses. Without effective measures for treatment and prevention, high rates of BLV infection can cause problems worldwide. BLV research is limited by the lack of a model system to assay infection. To overcome this, we previously developed the luminescence syncytium induction assay (LuSIA), a highly sensitive and objectively quantifiable method for visualizing BLV infectivity. In this study, we applied LuSIA for the high-throughput screening of drugs that could inhibit BLV infection. We screened 625 compounds from a chemical library using LuSIA and identified two that markedly inhibited BLV replication. We then tested the chemical derivatives of those two compounds and identified BSI-625 and -679 as potent inhibitors of BLV replication with low cytotoxicity. Interestingly, BSI-625 and -679 appeared to inhibit different steps of the BLV lifecycle. Thus, LuSIA was applied to successfully identify inhibitors of BLV replication and may be useful for the development of anti-BLV drugs.

## 1. Introduction

Like the human T-cell leukemia virus, bovine leukemia virus (BLV) belongs to the genus *Deltaretrovirus* of the family *Retroviridae*, and causes a lymphoproliferative disease characterized by B-cell lymphoma [1,2]. BLV infection also results in lower carcass weights and reduced milk production by infected cattle, leading to enormous economic losses in the dairy farming and livestock industry [3,4,5,6]. Currently, there are no effective measures for treating or preventing BLV infection; therefore, high infection rates are problematic worldwide [7]. To date, palliative measures have been taken, with limited success, including eliminating the infected cattle, segregating BLV-free animals, not using blood-contaminated devices, avoiding calf suckling, and exterminating hematophagous vector insects, such as horseflies [8,9]. Previously, a bovine major histocompatibility complex (BoLA) polymorphisms have been reported to be involved in the correlation between BLV infectivity and proviral load [10,11,12,13,14]. We recently reported that selective breeding of cattle carrying resistant *BoLA* alleles and culling of cattle carrying susceptible *BoLA* alleles can reduce the prevalence of BLV [11,15]. Notably, controlling BLV through an approach that takes advantage of the *BoLA* polymorphism is a long-term strategy for effective BLV eradication. Therefore, therapeutic, and preventive agents against BLV are needed for the control and eradication of BLV infection, and for protecting pedigrees that have industrial value [16,17].

Most anti-retroviral therapeutic agents have been developed for human immunodeficiency virus (HIV), and antiretroviral therapy (ART) has markedly improved the prognosis for patients infected with HIV [18,19,20]. ART can also be employed as a preventive measure against HIV transmission [21]. However, it has been reported that anti-HIV reverse transcriptase inhibitors are less effective against BLV [22]. Although anti-PD-1 antibodies [23] and violaceoid E [24] have been reported to suppress BLV replication, they have not been put to practical use. Thus, no effective anti-BLV drug has been approved to date.

Research into BLV is limited by the lack of an experimental system that can be used to assay BLV infection. We previously developed the luminescence syncytium induction assay (LuSIA) to remedy this. LuSIA exploits the indicator cells that form luminescent syncytia upon BLV infection and serves as a highly sensitive and objectively quantifiable method for visualizing BLV infectivity [25,26,27]. Using LuSIA, we confirmed that bovine CAT1 functions as a receptor for BLV infection [28]. We have also used LuSIA to detect BLV-infected cells in cows’ milk [29,30]. Therefore, LuSIA appears to be applicable for identifying potential anti-BLV drugs that can inhibit fluorescent syncytium formation.

In this study, we aimed to modify the LuSIA protocol to enable use for high-throughput screening, designate LuSIA-HTS, and screen 625 low-molecular-weight compounds from the RIKEN Natural Products Depository (NPDepo).

## 2. Materials and Methods

### 2.1. Cells and Virus

The CC81-GREMG cell line is derived from the mouse CC81 sarcoma virus-transformed feline cell line and expresses enhanced green fluorescent protein (EGFP) in a BLV Tax-inducible manner [26]. CC81-GREMG-CAT1 was established by stably transducing CC81-GREMG cells with the bovine CAT1 gene [27]. FLK-BLV is a sheep cell line with persistent BLV infection. Both cell lines were cultured with Dulbecco’s modified Eagle’s medium (DMEM) (Thermo Fisher Scientific, Waltham, MA, USA) supplemented with 10% fetal bovine serum (FBS; Sigma-Aldrich, St. Louis, MO, USA), as described previously [26,27]. To prepare infectious BLV particles, FLK-BLV cells (1 × 10^6^ cells) were cultured for 4 days, and the culture supernatant was collected, centrifuged at 440× *g* for 15 min to remove cell debris, and filtered through a 0.45 μm pore membrane (Merck Millipore, Darmstadt, Germany). The virus-containing supernatant was stored at −80 °C until use. For the binding analysis, virus particles were concentrated by centrifuging the culture supernatant at 42,200× *g* for 2 h at 4 °C, and the pellet was suspended in a 1/10 volume of fresh medium. The liquid sample containing the concentrated virus was aliquoted and stored at −80 °C until use.

### 2.2. Chemical Compounds

Overall, 400 and 225 screened chemical compounds were obtained for screening from the RIKEN chemical compounds pilot library and the similar compound collection library, respectively (NPDepo; http://www.npd.riken.jp/crdu/en/ accessed on 1 July 2022). Additional compounds BSI-601 (CAS#688037-94-7), -613 (CAS#688038-21-3), and -678 (CAS#676612-43-4) were purchased from Vitas-M Chemical Limited (Hong Kong, China), and BSI-625 (CAS#1212428-16-4), -677 (CAS#957000-43-0), and -679 (CAS#717836-99-2) were purchased from Pharmeks LTD. (Moscow, Russia), respectively. Dimethyl sulfoxide (DMSO), which was used as the solvent, and the negative control were purchased from Sigma-Aldrich.

### 2.3. Screening of Candidate Compounds

The LuSIA-HTS was used to screen chemical compounds for inhibitory activity against BLV infection. Thus, 1 × 10^4^ CC81-GREMG and 5 × 10^2^ FLK-BLV cells were suspended in 50 µL of culture liquid and seeded in 88 wells of a 96-well plate. At the same time, test compounds dissolved in DMSO (10 mg/mL) were added to 80 of those wells at a final concentration of 10 µg/mL. The same volume of DMSO alone was added to eight wells for use as a positive control for the maximal induction of fluorescent syncytium formation (100% control). In the remaining eight wells, CC81-GREMG cells and DMSO were added as a negative control to confirm the lack of fluorescent syncytium formation without BLV (0% control). Cells were incubated for 24 h at 37 °C under 5% CO_2_, washed with phosphate buffered saline (PBS), and fixed in 3.6% formaldehyde/PBS containing 10 µg/mL Hoechst 33342 (Sigma-Aldrich). Subsequently, EGFP-expressing fluorescent syncytia in each well were scanned using EVOS2 fluorescence microscopy (Thermo Fisher Scientific) with a four-fold objective lens in nine fields of view. Cells were counted using the HCS Studio Cell Analysis software (Thermo Fisher Scientific). Threshold values for fluorescence intensity and the size of the fluorescent area for defining the fluorescent syncytium were determined in each experiment based on the results of the 100% and 0% controls. The inhibition rate (IR) of each compound was calculated as follows:(1)$$\mathrm{IR}\;\left(\%\right)=\left[1-\left(\frac{\left(\#\mathrm{FSC}-\mathrm{average}\;\#\mathrm{FS}\_0\right)}{\left(\mathrm{avereage}\;\#\mathrm{FS}\_100-\mathrm{average}\;\#\mathrm{FS}\_0\right)}\right)\right]\times 100$$
where, #FSC represents the number of fluorescent syncytia (#FS) with an added test compound, and #FS_0 and #FS_100 represent the number of fluorescent syncytia in the 0 and 100% controls, respectively.

To assess the cytotoxicity of the compounds, the viability of treated cells was determined by the water-soluble tetrazolium salt-1 (WST-1) assay using the Premix WST-1 Cell Proliferation Assay System (TaKaRa Bio, Shiga, Japan), according to the manufacturer’s instructions. The optical density was then measured at a wavelength of 450 nm (OD450) using an EnSight plate reader (PerkinElmer, Waltham, MA, USA). Briefly, CC81-GREMG cells were seeded in each of 88 wells of a 96-well plate at 1 × 10^4^ cells and cultured for 24 h at 37 °C under 5% CO_2_. Then, compounds were added to 80 of those wells at a final concentration of 10 µg/mL, and the same volume of DMSO was added to eight wells for 100% cell viability control. A culture medium was added to the remaining eight empty wells, which did not contain cells, for the 0% cell viability control. Cell viability (CV) was calculated as follows:(2)$$\mathrm{CV}\;\left(\%\right)=\left(\frac{\left(\mathrm{OD}450C-\mathrm{average}\;\mathrm{OD}450\_0\right)}{\left(\mathrm{average}\;\mathrm{OD}450\_100-\mathrm{average}\;\mathrm{OD}450\_0\right)}\right)\times 100$$
where, OD450C represents the OD450 value measured by the WST-1 assay for cells treated with the test compound, and OD450_0 and OD450_100 represent the OD450 values of the 0 and 100% cell viability controls, respectively. 

Screened plates were validated by Z’ factors [31]. Z’ factor was calculated by the average and standard deviation (SD) of #FS and OD450 of controls, which required over 0.5, as follows:(3)$$\mathrm{LuSIA}\;Z^{\prime }\;\mathrm{factor}=1-\left(\frac{\left(3\times \mathrm{SD}\;\#\mathrm{FS}\_100\;\right)+\left(3\times \mathrm{SD}\;\#\mathrm{FS}\_0\right)}{\left(\mathrm{average}\;\#\mathrm{FS}\_100-\mathrm{average}\;\#\mathrm{FS}\_0\right)}\right)$$
(4)$$\mathrm{WST}1\;Z^{\prime }\;\mathrm{factor}=1-\left(\frac{\left(3\times \mathrm{SD}\;\mathrm{OD}450\_100\;\right)+\left(3\times \mathrm{SD}\;\mathrm{OD}450\_0\right)}{\left(\mathrm{average}\;\mathrm{OD}450\_100-\mathrm{average}\;\mathrm{OD}450\_0\right)}\right)$$

### 2.4. Dose-Response of Chemical Compounds

To assess the dose-response of BLV infection inhibitory activity, 500 µL of CC81-GREMG cells (2 × 10^5^ cells) were co-cultured with 500 µL of FLK-BLV cells (5 × 10^4^ cells), and 50.00, 16.67, 5.56, 1.85, 0.62, 0.21, 0.069, 0.023, 0.0076, and 0.0025 µg/mL of seed compounds in 12-well plate for 24 h at 37 °C under 5% CO_2_. CC81-GREMG and FLK-BLV cells co-cultured with DMSO (0% control), and CC81-GREMG cells only cultured with DMSO (100% control) were included in each plate. Following cultivation, the cells were washed with PBS and fixed in 3.6% formaldehyde/PBS containing 10 µg/mL Hoechst 33342 (Invitrogen, Carlsbad, CA, USA). Fluorescent syncytia were observed by fluorescence microscopy (BZ-9000, KEYENCE Co., Osaka, Japan) fitted with a 10-fold objective lens. Fluorescent syncytia were identified based on EGFP expression and gated by their area and intensity.

To assess the dose-response of compounds on cellular cytotoxicity, 50 µL of CC81-GREMG cells (2 × 10^4^ cells) were cultured with 50.00, 16.67, 5.56, 1.85, 0.62, 0.21, 0.069, 0.023, 0.0076, and 0.0025 µg/mL of seed compounds in 96-well plates for 24 h at 37 °C under 5% CO_2_. Cell viability was assessed by Cell-Titer Glo (Promega, Madison, WI, USA), according to the manufacturer’s instructions. Luminescence was measured by an Infinite 200 PRO multimode plate reader (TECAN, Männedorf, Switzerland).

The inhibitory concentration 50 (IC_50_) and cytotoxic concentration 50 (CC_50_) of seed compounds were calculated by approximation formulae using Microsoft Excel software version 2211. The selective index (SI) of seed compounds was also calculated as the CC_50_/IC_50_. Results were obtained from three independent experiments.

### 2.5. Time-of-Addition Assay

To assess time-dependent BLV infection inhibitory activity, 500 µL of CC81-GREMG cells (2 × 10^5^ cells) were co-cultured with 500 µL of FLK-BLV cells (5 × 10^4^ cells) in 12-well plates for 24 h at 37 °C in 5% CO_2_. Furthermore, 5 µg/mL of BSI-625 or 20 µg/mL of BSI-679 was supplemented at 0, 0.5, 2, 4, 8, 16, 20, and 24 h, respectively. CC81-GREMG and FLK-BLV co-cultured with DMSO were also supplemented using 100% control each time. After cultivation, the cells were washed with PBS and fixed in 3.6% formaldehyde/PBS containing 10 µg/mL Hoechst 33342. Fluorescent syncytia were observed and analyzed as described above.

### 2.6. Quantification of Enhanced-Green Fluorescence Protein mRNA

CC81-GREMG cells (1 × 10^6^ cells) were co-cultured with FLK-BLV cells (2.5 × 10^5^ cells) in 5 cm dishes for 24 h at 37 °C under 5% CO_2_, supplemented with 20 µg/mL of BSI-625 or -679, respectively. Co-cultures of CC81-GREMG and FLK-BLV cells with DMSO, and cultures of CC81-GREMG cells only were used for the 100 and 0% control, respectively. After cultivation, the cells were washed with PBS and total RNA was extracted using a RNeasy Plus Mini kit (QIAGEN, Hilden, Germany), according to the manufacturer’s instructions. Complementary DNA was reverse-transcribed from 1 µg of total RNA using ReverTra ACE (TOYOBO, Osaka, Japan), according to the manufacturer’s instructions. Reverse-transcribed mRNA of EGFP and feline beta-2-microglobulin (B2M; house-keeping gene as the feline cell internal control) [32] were measured by real-time PCR using Quant Studio 3 (Thermo Fisher Scientific) with specific primer pairs and TUNDERBIRD NEXT SYBR qPCR Mix (TOYOBO). PCR was performed under the following conditions: pre-incubation at 95 °C for 1 min; thereafter, 95 °C for 15 s and 60 °C for 60 s, then repeated for 40 cycles. Specific primers were synthesized by Integrated DNA Technologies Inc. (IDT: Coralville, IA, USA) and their sequences were as follows: EGFP forward: ACGACGGCAACTACAAGAC, reverse: TGCTTGTCGGCCATGATATAG, B2M forward: TTGTGGTCTTGGTCCTGCTCG, reverse: TTCTCTGCTGGGTGACGGGA [32]. Relative EGFP mRNA expression was normalized and quantified using the ΔΔCt method compared to feline B2M expression [33]. Results are presented as the means and standard deviation of three independent experiments.

### 2.7. Quantification of Reverse-Transcribed Viral DNA

CC81-GREMG-CAT1 cells (1.2 × 10^6^ cells) were seeded in six-well plates and incubated overnight at 37 °C under 5% CO_2_. Cells were inoculated with 1 mL FLK-BLV culture supernatant containing infectious BLV particles with or without 20 μg/mL BSI-625 or BSI-679 at 37 °C under 5% CO_2_ for 1 h, with rocking every 15 min. Cells were then washed twice with DMEM. Fresh 10% FBS/DMEM with or without 20 μg/mL of BSI-625 or BSI-679 was added, and cells were cultured at 37 °C under 5% CO_2_ for an additional 5 h. Whole cellular DNA was extracted using the Wizard genomic DNA purification kit (Promega), according to the manufacturer’s instructions. Viral genomic RNA was disrupted by 30 min treatment with RNase A. BLV-*tax* and cat genomic DNA was analyzed by real-time PCR with specific primer pairs and TUNDERBIRD NEXT SYBR qPCR Mix. The PCR conditions were the same as those described previously. Specific primers synthesized by IDT were as follows: BLV-*tax* forward: TGGAACAACTTAGTAACGCATC, reverse: GCTCGCCTAGGGGTAGAATAC, feline genomic-*gapdh* forward: GACCACTTTGTCAAGCTCATTTC, reverse: GATCACGAGTTCAGGCCTATTT. The amount of reverse-transcribed viral DNA was normalized and quantified using the ΔΔCt method with feline genomic-*gapdh*. The results are presented as the mean and standard deviation of three independent experiments.

### 2.8. Viral Particle Binding Assay

Two-hundred-microliters of CC81-GREMG cells (1 × 10^6^ cells) or 10-fold concentrated BLV-containing supernatant were supplemented with 20 µg/mL of BSI-625, -679, or DMSO, and allowed to stand for 20 min at 25 °C. BSI-625- or -679-treated cells were mixed with DMSO-treated supernatant, and DMSO-treated cells were mixed with BSI-625- or -679-treated supernatant. DMSO-treated cells mixed with DMSO-treated supernatant were used as the 100% control, and cells mixed with the medium were used as the 0% control. The mixture was incubated at 4 °C for 2 h, and cells were washed five times with cold PBS. Thereafter, cell-surface bound virus was stained with anti-BLV monoclonal antibody (mAb) (BLV-1: VRMD, Pullman, WA, USA) and Alexa-647 conjugated anti-mouse mAb (Invitrogen). Cell-surface fluorescence was measured by BD LSR X-20 (BD Bioscience, San Jose, CA, USA) and the mean fluorescence intensity of cells was compared with that of DMSO-treated controls.

## 3. Results

### 3.1. First Screening of BLV Replication Inhibitors Using LuSIA-HTS

Previously, we reported that the LuSIA can easily and objectively measure BLV infectivity [25,26,27]. Here, we aimed to identify inhibitors of BLV infection using LuSIA with a small compound library. The LuSIA was adapted to allow use for high-throughput screening for inhibitors in 96-well plates, as LuSIA-HTS. In addition, the WST-1 assay was used to assess the toxicity of screening compounds. The screening scheme is shown in Figure 1.

First, we screened small compounds belonging to the pilot library, and 400 compounds with representative structures were selected from all compounds deposited in NPDepo. These compounds were adjusted to 10 μg/mL and dispensed into five 96-well plates, each containing 80 compounds. Each plate had 8 wells of DMSO-treated control and 8 wells of non-infected control. To validate our assay, the Z’ factors were calculated for each assay plate. All that exceeded the 0.5 threshold were validated (Figure 2A). LuSIA-HTS revealed that the formation of fluorescent syncytium decreased by over 50% (by 54) among the 400 compounds, which included the cellular toxicity of treated compounds (Figure 2B). Overall, 148 compounds resulted in cell viability that exceeded 80% (Figure 2B). Next, we determined the criteria of the hit compounds reaching >50% inhibition of infectivity rate and >90% of cell viability and selected the top three compounds as hit compounds from the pilot library. BSI-066, -206, and -234 demonstrated 90.7, 53.4, and 71.9% inhibitory activity, and 91.9, 94.1, and 91.9% viability, respectively. Depletion of fluorescent syncytium and chemical structures are shown in Figure 2C,D, respectively.

### 3.2. Second Screening of BLV Replication Inhibitors with Similar Structures to the Hits from the First Screening

Following the first screening using LuSIA-HTS, the top three compounds (BSI-066, -206, and -234) were successfully identified from 400 compounds in the pilot library. The pilot library was constructed using 400 representative structures of approximately 25,000 compounds registered with NPDepo. Furthermore, we searched for similar compounds among all compounds registered in NPDepo to confirm their activity (Figure 1). BSI-066 or biochanin A, one of the isoflavones, has a simple structure, and over 200 similar compounds are stored in NPDepo. Thus, we have selected the top 198 compounds similar to BSI-066 based on their similarity (Table 1 and Appendix A). Five and 22 similar compounds to BSI-206 and -234 were identified in NPDepo, respectively (Appendix A). In the second screening, 225 similar compounds were selected and their inhibitory activity against BLV infection and cellular toxicity were assayed per the first screening. Only BSI-526 presented a >50% infectivity inhibition rate and >90% cell viability out of 198 compounds that were similar to BSI-066 (Figure 3A, Table 1 and Appendix A). Unfortunately, there was no clear correlation between structure and activity among the 198 compounds similar to BSI-066. Conversely, two of five compounds similar to BSI-206 (BSI-623 and -625) and three of 22 compounds similar to BSI-234 (BSI-601, -611, and -613) presented strong inhibitory activity without toxicity (Figure 3B, Appendix A). Thus, both the BSI-206 and -234 structures demonstrated inhibitory activity; therefore, we analyzed their mechanism of action and excluded the BSI-066 structure compounds from further experiments.

Compounds with similar structures of BSI-206 and -234 and expected to have high inhibitory activity and low toxicity based on secondary screening results (Appendix A) were searched from existing compounds available for purchase to compare inhibitory activity and survival rates, respectively. We identified two commercially available compounds that were similar to the BSI-206 structure (BSI-625 and -677) and four commercially available compounds that were similar to the BSI-234 structure (BSI-601, -613, -678, and -679) (Table 1). 

### 3.3. Identification of Two Final Hits with Selected Structures in the Second Screening

These existing similar compounds available for purchase were obtained and evaluated for inhibitory activity and toxicity in the presence of 50.00, 16.67, 5.56, 1.85, 0.62, 0.21, 0.069, 0.023, 0.0076, and 0.0025 µg/mL, respectively (Figure 1). All similar compounds presented inhibitory activity at lower concentrations and with lower toxicity (Figure 4A,B). The lowest concentrations of diluted BSI-625, BSI-677, BSI-601, BSI-613, BSI-678, and BSI-679 that passed the criteria, as reported for the first and second screenings (inhibition rate >50% and viability >90%), were 1.86, 5.56, 16.7, 16.7, 16.7, and 5.56 µg/mL, respectively.

Thus, we selected BSI-625 as a similar compound to BSI-623 and BSI-679 as a similar compound to BSI-613, due to their inhibition activity at low concentrations. To characterize these two structural inhibitor candidates, we calculated their IC_50_, CC_50_, and SI (Figure 5 and Table 2). For BSI-625, the IC_50_ was 1.6 ± 0.8 µg/mL, the CC_50_ was 32.0 ± 11.9 µg/mL, and the SI was at 20.1. For BSI-679, these values were 6.3 ± 1.5 µg/mL, 43.8 ± 3.0 µg/mL, and 6.9, respectively.

### 3.4. Time-of-Addition Assay for the Two Final Hits

Next, to determine the point at which these two inhibitor candidates acted during the BLV lifecycle, the compounds were added 0, 0.5, 1, 2, 4, 8, 16, 20, and 24 h post co-cultivation of CC81-GREMG and FLK-BLV cells. Both BSI-625 and -679 were found to be most effective when added 0 to 0.5 h after co-cultivation (Figure 6A,B). Furthermore, the inhibitory effect of BSI-625 remained at 16 h post co-cultivation; however, that of BSI-679 was observed within 4 h of co-cultivation. This indicated that BSI-625 and -679 function at different stages of the BLV lifecycle and exert alternative mechanisms on the inhibition of BLV replication.

### 3.5. Effect of the Two Final Hit Compound Inhibitors on the Adsorption of BLV Particles to the Target Cell Surface

Both BSI-625 and -679 were most effective after 0–0.5 h of co-cultivation. This suggested that the two compounds act at an early stage of the viral lifecycle rather than at a later stage. For example, BLV direct contact to the cellular surface protein CAT1, as a cell surface receptor for BLV infection during the adsorption stage. Therefore, we investigated the effect of the compound on the adsorption of BLV particles on the surface of CC81-GREMG cells. First, CC81-GREMG cells were incubated with 20 µg/mL BSI-625, -679 or DMSO for 20 min at 25 °C, and then with 10-fold concentrated FLK-BLV supernatants, including infectious viral particles, for 2 h at 4 °C. The amount of BLV particles bound to the cell surface was determined by flow cytometry using anti-Env mAb (BLV-1). Flow cytometric analysis revealed that neither BSI-625 nor -679 affected viral adsorption on the cell surface when 20 µg/mL compounds were added to target cells ahead of co-cultivation (Figure 7A). Furthermore, 10-fold concentrated FLK-BLV supernatant was incubated in the presence of 20 µg/mL BSI-625, -679, or DMSO at 25 °C for 20 min and then reacted with CC81-GREMG cells for 2 h at 4 °C. As shown in Figure 7B, binding between CC81-GREMG cells and BLV particles was not affected by either BSI-625 or -679. Our results indicated that BSI-625 and -679 affected BLV replication at a step after viral adsorption.

### 3.6. Effect of the Two Final Hit Compounds on Reverse-Transcribed Viral DNA

Reverse transcription after adsorption step is likely affected by both compounds. Next, we assessed whether the two compounds affected the amount of viral cDNA. CC81-GREMG-CAT1 cells, which overexpress the BLV receptor CAT1 [27,28], were inoculated with cultured supernatant from FLK-BLV cells, including infectious BLV viral particles, in the presence of compounds for 1 h at 37 °C. After inoculation, cells were cultured with compounds for an additional 5 h, after which whole cellular DNA was extracted. Reverse-transcribed viral DNA was measured by real-time PCR using BLV *tax*-specific primers and normalized to the amount of the feline genomic DNA. There was a significant decrease in the amount of viral cDNA extracted from BSI-679-treated cells, which decreased by as much as 75% compared to the DMSO-treated control (Figure 8). Conversely, there was no change in the amount of viral cDNA extracted from BSI-625-treated cells. These results indicated that BSI-679 affected a step before reverse transcription after adsorption, while BSI-625 may act during a stage later than reverse transcription in the lifecycle of BLV.

### 3.7. Enhanced-Green Fluorescence Protein mRNA Expression by the Two Final Hits

Finally, we evaluated the effect of these compounds on the late phase of the BLV lifecycle. CC81-GREMG cells carry the BLV LTR-U3 region and express EGFP upon exposure to the BLV transactivating protein Tax. Therefore, to confirm whether these compounds inhibit BLV infection at a stage between viral mRNA transcription and integration, we assessed EGFP mRNA expression in BLV-infected CC81-GREMG cells following treatment with the compounds. Both BSI-625 and -679 decreased the mRNA expression of EGFP, which is transcribed by the viral transcription factor Tax (Figure 9). This result indicated that these candidate inhibitors function before viral RNA transcription.

## 4. Discussion

Previously, we developed a highly sensitive method to analyze the infectivity of BLV and named it the luminescence syncytium induction assay (LuSIA) [25,26,27]. LuSIA provides an objective method for quantifying BLV infectivity and has been used to demonstrate the presence of infected cells in milk as well as the peripheral blood of infected cows [29,30,34]. We thought that if we could find a drug that could reduce BLV infection using LuSIA, it may suppress BLV transmission in cattle. Here, we constructed a high-throughput screening system using LuSIA to search for new candidate drug compounds with the ability to inhibit BLV infection. Using LuSIA-HTS and the WST-1 assay to measure cytotoxicity, we selected the top three compounds (BSI-066, BSI-206, and BSI-234) with low toxicity and high BLV replication inhibitory activity among 400 compounds from the NPDepo pilot library. We successfully identified BSI-625, which is similar to the structure of BSI-206, and BSI-679, which is similar to BSI-234 as inhibitors of BLV replication. However, we were unable to obtain inhibitors for compounds with structures similar to BSI-066.

One compound, BSI-066, is an isoflavone named biochanin A, which has been reported to possess phytoestrogenic activity [35] and regulate inflammation [36,37]. Isoflavones have been reported to suppress cancer growth [38,39] and inhibit some viral infections as detailed below. Biochanin A and baicalein inhibit the replication of influenza H5N1 [40,41]. Genistein, a compound similar to biochanin A, has been reported to inhibit the replication of the African swine fever virus [42]. A selective estrogen receptor modulator (SERM) has been shown to inhibit dengue virus [43] and Ebola virus [44] infections, suggesting that estrogen-like activity may also affect BLV infection. In a preliminary investigation, we examined the BLV inhibitory activity of estrogens, estrogen receptor inhibitors, and SERMs; however, none of these compounds were found to exert inhibitory activity [45]. NPDepo stocks many flavonoids and isoflavonoids, including biochanin A analogues. In this study, the top 198 analogues with high similarity to biochanin A were used to screen for inhibitory activity. Consequently, several compounds were found to possess some inhibitory activity, including the BSI-526 (flavone), while conversely, several analogs tended to promote the formation of the syncytium, including BSI-484 (kaempferol). Since there was no correlation between the structure and inhibitory activity of these analogs, and no compounds presented significant efficacy [45], the mechanism of BLV replication inhibition by biochanin A could not be elucidated in this study.

Several analogs of another compound, BSI-206, which carries an allantolactone skeleton, demonstrated high inhibitory activity against BLV replication, resulting in the highly active compound BSI-625, which is effective with low toxicity and at low concentrations. BSI-625 inhibited EGFP mRNA transcription by the BLV viral transcription factor Tax, while reverse transcription of viral genomic DNA was not inhibited. This suggested that BSI-625 acts between the nuclear transfer of genomic DNA and viral gene transcription in the BLV lifecycle. Indeed, BSI-625 was most effective between 0 and 0.5 h of co-cultivation, and its inhibitory effect remained at 16 h post co-cultivation. BSI-625 and compounds with analogous skeletal structures have been reported as potential anticancer agents that inhibit mutant K-Ras-related signaling [46]. To date, no relationship between BLV and mutant K-Ras has been reported; however, Tax or G4 of BLV enhance transformation by Ha-Ras [47,48]. It has also been suggested that Ras suppresses apoptosis and regulates HTLV-1 transformation [48,49,50]. Further studies are needed to define the target of BSI-625.

Many analogs of the remaining compound, BSI-234, which has a biindole skeleton, were shown to inhibit BLV replication. BSI-679, a highly potent drug candidate with the same structure, resulted in inhibition when added up to 2 h after co-culture, suggesting that it acts early during the BLV lifecycle. In addition to EGFP mRNA transcription by Tax, BSI-679 also inhibited reverse transcription of viral genomic DNA, suggesting that it acts at a stage before genomic DNA reverse transcription. However, BSI-679 did not inhibit the adsorption of viral particles on the cell surface, suggesting that this compound functions at a step between membrane fusion and reverse transcription. Furthermore, since it strongly inhibits syncytium formation in the LuSIA, inhibition of membrane fusion is strongly suspected. Interestingly, the biindole skeleton of BSI-679 resembles that of sisunatovir (RV-521) [51], a drug that binds to the F-protein of respiratory syncytial virus and inhibits membrane fusion. Therefore, further analyses are needed to determine the target of BSI-679 and its mechanism of action.

The present study also demonstrated that two different BSI-206 and -234 structures, but not the BSI-066 structure, possessed inhibitory activity. Indeed, BSI-625 carrying the BSI-206 structure and BSI-679 carrying the BSI-234 structure were identified as novel antiviral compounds with anti-BLV activity. Interestingly, BSI-625 and -679 act at different stages of the lifecycle and exert alternative mechanisms of action on the inhibition of BLV replication: BSI-679 was functional at a step before reverse-transcription after adsorption, and BSI-625 inhibited EGFP mRNA transcription by the BLV viral transcription factor Tax. Although we expect that the effects of BSI-679 and -625 would be useful for reducing the risks related to the transmission and development of EBL, it is unclear whether the two compounds will exert an effect in vivo. Therefore, further studies based on the development of the BSI-206 and -234 structures will be useful for the development of anti-BLV drugs.

In this study, we constructed a LuSIA-based screening method to identify candidate drugs that inhibit BLV replication with two distinct targets. The two candidate compounds and their analogues may contribute to the development of therapeutic agents for BLV infection.

## Figures and Tables

**Figure 1 viruses-15-00004-f001:**
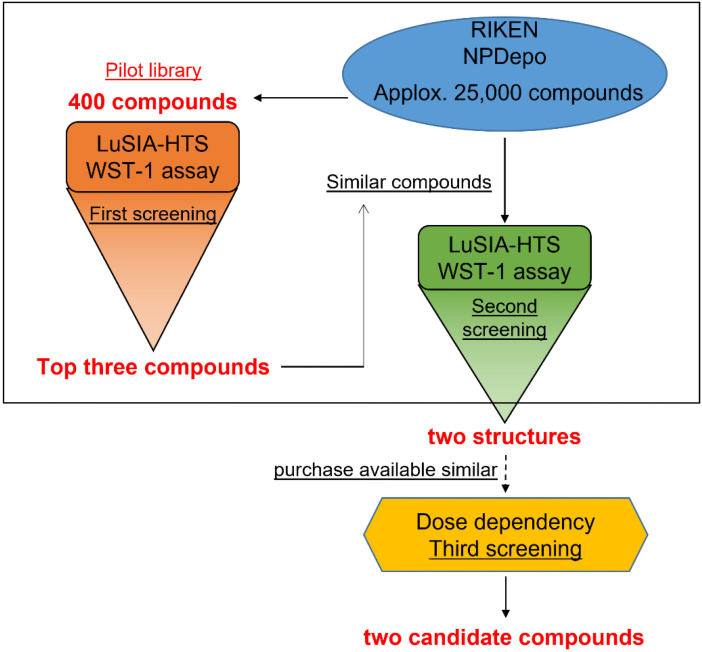
Screening scheme. Schematic presentation of bovine leukemia virus (BLV) inhibitor screening. The first screening employed the pilot library, which included 400 compounds of typical structure from 25,000 compounds in RIKEN Natural Products Depository (NPDepo). The screening was performed by luminescence syncytium induction assay high-throughput screening (LuSIA-HTS) and water-soluble tetrazolium salt-1 (WST-1) assay. Following the first screening, the NPDepo database was searched for similar structures of the hit compounds. Similar compounds were screened as reported for the first screening to identify strong inhibitor candidates.

**Figure 2 viruses-15-00004-f002:**
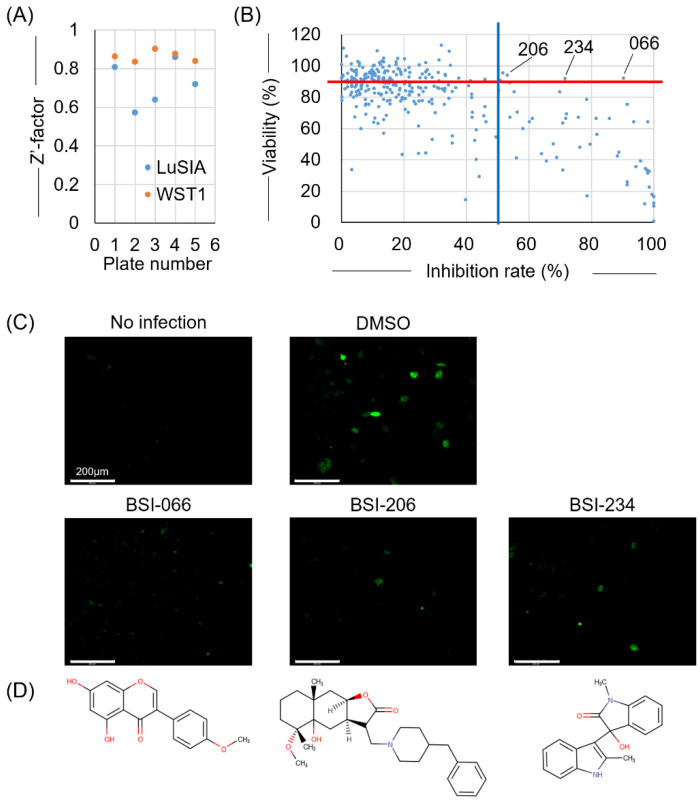
Result of the first screening. To select the inhibitor candidates from NPDepo, the viral inhibitory activity of compounds was screened using LuSIA-HTS and their toxicity was screened using WST-1 assay. LuSIA-HTS was performed using 10^4^ cells of CC81-GREMG and 5 × 10^2^ cells of FLK-BLV cells co-cultured with 10 μg/mL candidate compounds for 24 h at 37 °C, 5% CO_2_. After 24 h, formed-fluorescence syncytia were counted, and the inhibition rate was calculated. The WST-1 assay was performed under the same conditions except for the use of FLK-BLV cells. Cell viability was assessed by Premix WST-1 Cell Proliferation Assay System. (**A**) Screening assays validated with Z’ factors. (**B**) Plots showing the inhibition rate and cell viability of 400 compounds in the pilot library from NPDepo. Hit criteria are defined as >50% infectivity inhibition rate (blue line) and >90% of cell viability (red line). (**C**) Fluorescent syncytium formation treated with hit compounds obtained from the first screening. (**D**) Chemical structure of hit compounds obtained from the first screening. Scale bar = 200 µm.

**Figure 3 viruses-15-00004-f003:**
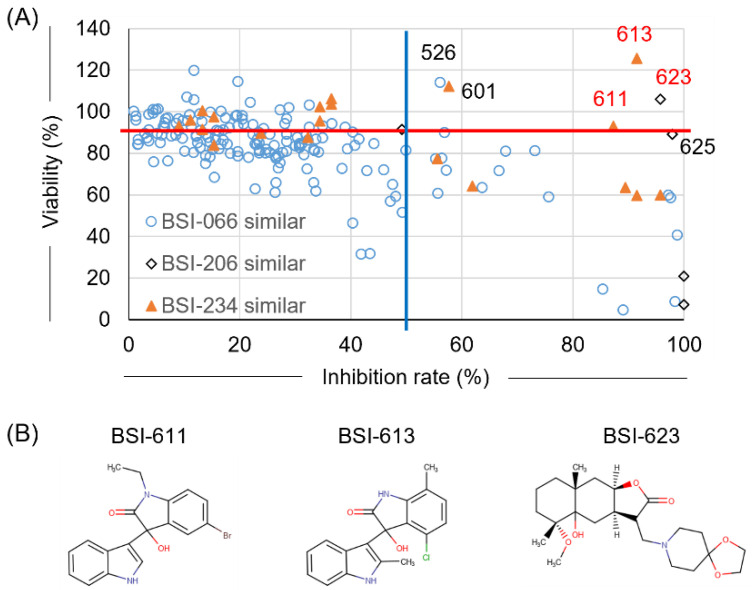
Result of the second screening. To identify more effective compounds compared with the first screening, similar compounds selected from all 25,000 compounds registered in NPDepo were assessed for BLV inhibitory activity using LuSIA-HTS, and toxicity was assessed using the WST-1 assay. Assays were performed under the same conditions as described for the first screening. (**A**) Plot showing the inhibition rate and cell viability of 225 similar compounds that were selected from NPDepo. Hit criteria were defined as >50% infectivity inhibition rate (blue line) and >90% cell viability (red line). (**B**) Structure of hit compounds obtained from the second screening.

**Figure 4 viruses-15-00004-f004:**
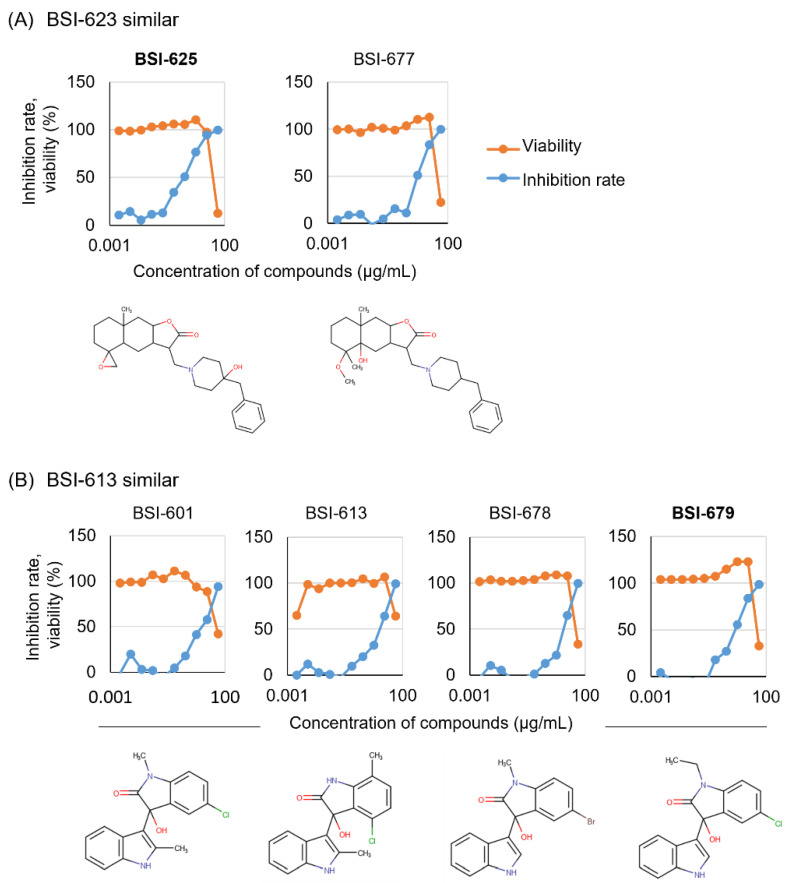
Dose-dependency of the six candidates with two selected structures was obtained from the second screening. Six purchased similar compounds were used to assess BLV inhibition activity using the LuSIA-HTS, and to assess toxicity using the WST-1 assay. Serially diluted compounds (50.00, 16.67, 5.56, 1.85, 0.62, 0.21, 0.069, 0.023, 0.0076, and 0.0025 µg/mL) were cultured with 2 × 10^5^ CC81-GREMG cells and 10^5^ FLK-BLV cells in 12-well plate for 24 h. After 24 h, formed fluorescent syncytia were counted and the inhibition rate was calculated. The WST-1 assay was performed under the same conditions except for the use of FLK-BLV cells in a 96-well plate. Cell viability was assessed using a Premix WST-1 Cell Proliferation Assay System. The results and structures of similar compounds of BSI-623 (**A**) and BSI-613 (**B**) are presented.

**Figure 5 viruses-15-00004-f005:**
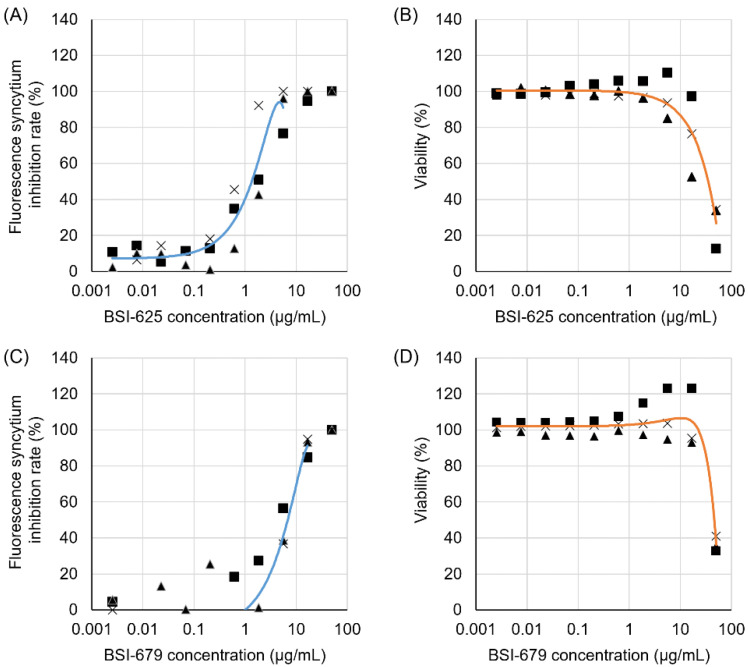
Inhibitory activity and cell toxicity of the final hit compounds, BSI-625 and -679. Inhibition rates and viability of BSI-625 and -679 were assessed in the absence or presence of serially diluted compounds (50.00, 16.67, 5.56, 1.85, 0.62, 0.21, 0.069, 0.023, 0.0076, and 0.0025 µg/mL) by the LuSIA and WST-1 assay. Serially diluted compounds were cultured with 2 × 10^5^ CC81-GREMG cells and 10^5^ FLK-BLV cells in 12-well plates for 24 h. After 24 h, formed-fluorescent syncytia were counted, and the inhibition rate was calculated. The inhibition rates for BSI-625 (**A**) and BSI-679 (**C**) are shown, respectively. The WST-1 assay was performed under the same conditions, except for the use of FLK-BLV cells in 96-well plates. Cell viability was assessed using the Premix WST-1 Cell Proliferation Assay System. The results for BSI-625 (**B**) and BSI-679 (**D**) are shown. The results show three independent experiments (results showed as triangle, square and cross, respectively), and an approximate curve calculated using the average of three experiments. Calculated inhibitory concentration 50, cytotoxic concentration 50, and their selective indices are indicated in Table 2.

**Figure 6 viruses-15-00004-f006:**
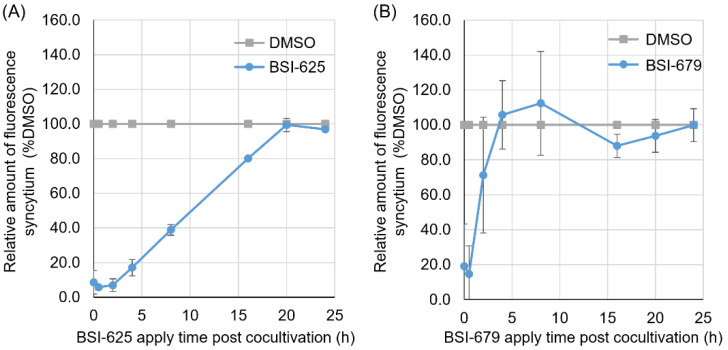
Time-of-addition assay for the final hit compounds BSI-625 and -679. CC81-GREMG cells and FLK-BLV cells were co-cultured for 24 h, and either BSI-625 or BSI-679 was added 0, 0.5, 2, 4, 8, 12, 16, 20, and 24 h post-co-culture. After 24 h of co-culture, the number of fluorescent syncytia in 25 fields was counted. The relative amounts of fluorescent syncytium were normalized to the DMSO-treated cells (100% control). The results are presented as the mean and standard deviation of three independent experiments for BSI-625 (**A**) and BSI-679 (**B**).

**Figure 7 viruses-15-00004-f007:**
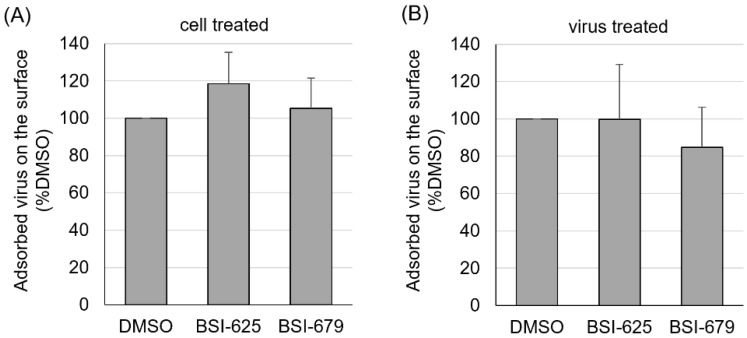
Viral particle adsorption on the cell surface for the final hit compounds BSI-625 and -679. (**A**) CC81-GREMG cells were cultured with 20 µg/mL of BSI-625, -679, or DMSO for 20 min at 25 °C and then incubated with 10-fold concentrated cultured FLK-BLV supernatant, including viral particles, for 2 h at 4 °C. (**B**) The 10-fold concentrated cultured FLK-BLV supernatants, including viral particles, were incubated with 20 µg/mL of BSI-625, -679, or DMSO for 20 min at 25 °C, and incubated with CC81-GREMG cells for 2 h at 4 °C. DMSO-treated samples were used as the 100% control and CC81-GREMG cells mixed with medium without viral particles were used as 0% control. Post incubation, cells were washed five times with cold-PBS and stained with anti-Env mAb followed by Alexa-647-conjugated anti-mouse mAb. Fluorescence on the cell surface was measured by flow cytometry and the mean fluorescence intensities were compared.

**Figure 8 viruses-15-00004-f008:**
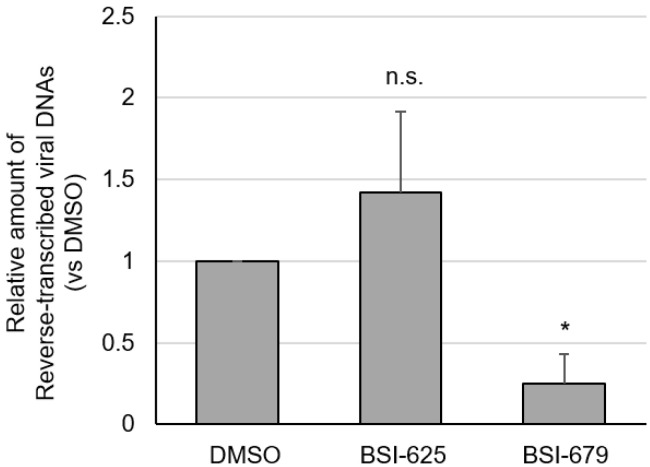
The amount of reverse-transcribed viral DNA in BLV infected CC81-GREMG-CAT1 cells in the presence of BSI-625 and -679. CC81-GREMG-CAT1 cells were inoculated with cultured supernatant from FLK-BLV cells containing BLV particles in the presence of 20 μg/mL of BSI-625, -679, or DMSO for 1 h, with rocking for 15 min. Cells were washed twice with the medium and fresh medium was then added in the presence of 20 μg/mL of BSI-625, -679 or DMSO, and cultured for a further 5 h. Whole cellular DNA was extracted using Wizard genomic DNA purification kit (Promega). BLV-*tax* and cat genomic DNA was measured in real-time-PCR and the amount of reverse-transcribed viral genomic DNA was calculated. The results show the mean and standard deviation of three independent experiments. *p*-values were calculated and * *p* < 0.05 was considered a statistically significant difference. n.s. means not significant.

**Figure 9 viruses-15-00004-f009:**
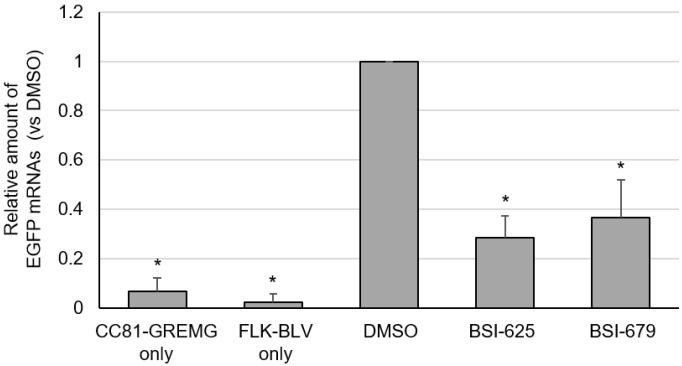
The mRNA expression of enhanced green fluorescent protein (EGFP) in co-cultures of CC81-GREMG and FLK-BLV cells for the final hit compounds BSI-625 and -679. CC81-GREMG and FLK-BLV cells were co-cultured in the presence of BSI-625, -679, or DMSO for 24 h. Total RNA was extracted by RNeasy Plus Mini kit (QIAGEN). Complementary DNA was synthesized from total RNA using ReverTra-ACE (TOYOBO). EGFP and catB2M mRNA were amplified by real-time PCR and relative EGFP mRNA expression was determined using the ΔΔCt method. The results are presented as the mean and standard deviation of three independent experiments. *p*-values were calculated and * *p* < 0.05 was considered a statistically significant difference.

**Table 1 viruses-15-00004-t001:** Summary of second screening.

First Screening Top Three Compounds	Second Screening		
	Number of Similar Compounds	Top Hit Compounds	Available Candidate of Next Screening
BSI-066	198	BSI-526	exit *
BSI-206	5	BSI-623	BSI-625 BSI-677 as commercially available similar compound
BSI-234	22	BSI-613	BSI-601 and -613 BSI-678 and -679 as commercially available similar compounds

* We could not find a commonly used structure showing sufficiently high activity, so it was excluded from the next experiment.

**Table 2 viruses-15-00004-t002:** Characterizations of seed compounds.

Compound	IUPAC Name	Screened Similar Compound	IC_50_ (µg/mL)	CC_50_ (µg/mL)	SI
BSI-625	(3aR,8aR,9aR)-3-((4-benzyl-4-hydroxypiperidin-1-yl)methyl)-8a-methyl-decahydro-2*H*-spiro(naphtho (2,3-b)furan-5,2’-oxiran)-2-one	similar compound of BSI-206 and -623	1.6 ± 0.8	32.0 ± 11.9	20.1
BSI-679	5-chloro-1-ethyl-3-hydroxy-2,3-dihydro-1*H*,1’*H*-(3,3’-biindol)-2-one	similar compound of BSI-234 and -613	6.3 ± 1.5	43.8 ± 3.0	6.9

## Data Availability

All data analyzed for the purposes of this manuscript are included in this article.

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
