# Peer review of "Application of the Luminescence Syncytium Induction Assay to Identify Chemical Compounds That Inhibit Bovine Leukemia Virus Replication"

_viruses, 2022, doi:10.3390/v15010004_

Round 1
Reviewer 1 Report
Review report
Title : Application of the luminescence syncytium induction assay to identify chemical compounds that inhibit bovine leukemia virus replication
The manuscript describes the use of a luminescence syncytium induction assay (LuSIA), previously developed by the authors, to test for Bovine leukemia virus (BLV) inhibitory activity, in a high throughput manner. At first, they performed an initial screening of 400 compounds, finding 3 of them with inhibitory activity and absence of citotoxicity. Based on the chemical structure of these 3 compounds, they carried out a second screening and finally found two different structures with inhibitory activity. Finally, they performed experiments in order to determine the step of the viral cycle in which interference occurs.
Although there is a long way from the bench (in vitro inhibitory activity) towards the use of any of these drugs in vivo, to control BLV infection, these findings may contribute to the development of therapeutic agents for BLV infection.
The manuscript is easy to follow and, in general, well written. However I have made minor observations, which must be answered before publication.
Minor comments :
1. Abstract, line 28 : 607 compounds were tested, not 627 (400 in initial screening and 207 in second screening). Please check
2. Page 2, line 46: please replace the words « breast feeding », which is not usually employed in the case of cattle
3. Introduction, lines 47 to 50 : findings on the association of BLV proviral load and polymorphism at BoLA are not recent. This sentence should be re-written as it is not gramatically correct. Moreover, selected references (10-15) are not representative of the main papers contributing to this idea, but inappropriate self citations.
4. Material and methods, line 92 : 207 compounds were tested, not 227. Besides, it is not possible to check for the compounds tested by the link provided. Please, give any way to have access to the list of compounds tested (for example as a supplementary excell file).
5. Page 3, line 139 : please replace « standard derivative » for « standard deviation »
6. Page 6 : figure 1 is lacking
7. Table 1 : the sentence about compounds related to BSI-066 at the end of the table has no sense and should be re-written.
8. Page 10, line 342, legend to figure 5: please check the letters between parenthesis
9. Discussion, line 451 : last sentence of the paragraph shpuld be revised (the expression : « but not BSI-066 » is inadequately located.
10. Page 13, line 454 : replace « as below » for « as detailed below ».
11. Page 14, lines 462 to 464 : please revise the construction of this sentence.
12. Line 482 : replace « target BSI-625 » for « target of BSI-625 »
13. Line 496: please check an error of typing.
14. Reference 23 has errors, please check.
Author Response
Response to reviewer 1:
Thank you for your gently review and comments. We appreciate your comments and have amended our manuscript accordingly.
Reviewer 1 comment
Comments and Suggestions for Authors
Review report
Title : Application of the luminescence syncytium induction assay to identify chemical compounds that inhibit bovine leukemia virus replication
The manuscript describes the use of a luminescence syncytium induction assay (LuSIA), previously developed by the authors, to test for Bovine leukemia virus (BLV) inhibitory activity, in a high throughput manner. At first, they performed an initial screening of 400 compounds, finding 3 of them with inhibitory activity and absence of citotoxicity. Based on the chemical structure of these 3 compounds, they carried out a second screening and finally found two different structures with inhibitory activity. Finally, they performed experiments in order to determine the step of the viral cycle in which interference occurs.
Although there is a long way from the bench (in vitro inhibitory activity) towards the use of any of these drugs in vivo, to control BLV infection, these findings may contribute to the development of therapeutic agents for BLV infection.
The manuscript is easy to follow and, in general, well written. However I have made minor observations, which must be answered before publication.
Minor comments :
- Abstract, line 28 : 607 compounds were tested, not 627 (400 in initial screening and 207 in second screening). Please check
Responses:
Thank you for your kind advice. We apologize for the error in the total number of compounds. After checking all the compound counts, we found another mistake in the number of similar compounds for BSI-066. Apologies for the mistake and thanks for the correction. Throughout the manuscript, the number of BSI-066 analogues was corrected from 180 to 198, the total number of analogues from 227 or 207 to 225, and the total number of compounds from 627 to 625. Here are the changes and their locations:
Corrected from
Line 28: 627 to 625
Line 73: 627 to 625
Line 93: 227 to 225
Line 285: 180 to 198
Line 288: 207 to 225
Line 290: 180 to 198
Line 292: 180 to 198
Line:310: 207 to 225
Line 467: 180 to 198
Table 1: 180 to 198
- Page 2, line 46: please replace the words « breast feeding », which is not usually employed in the case of cattle
Responses:
Thank you for your kindly advise. According to your recommendation, we replace the words “breast feeding” to “calf suckling”.
Line 46: “breast feeding” to “calf suckling”
- Introduction, lines 47 to 50 : findings on the association of BLV proviral load and polymorphism at BoLA are not recent. This sentence should be re-written as it is not gramatically correct. Moreover, selected references (10-15) are not representative of the main papers contributing to this idea, but inappropriate self citations.
Responses:
Thank you for your kindly advise. We rewritten the lines 47 to 50 as below. We also cited recent major papers from around the world and limited our self-citations to important ones.
Line 47: ” Recently, a bovine major histocompatibility complex (BoLA) polymorphism associated with BLV infectivity and proviral load and make it possible to reduce BLV prevalence by the selective breeding of cattle carrying the resistant BoLA allele and culling of cattle carrying the susceptible BoLA allele [10-15].” to
“Previously, a bovine major histocompatibility complex (BoLA) polymorphisms have been reported to be involved in the correlation between BLV infectivity and proviral load [10-14]. We recently reported that selective breeding of cattle carrying resistant BoLA alleles and culling of cattle carrying susceptible BoLA alleles can reduce the prevalence of BLV [11,15].”
Citations:
- LaHuis, C.H.; Benitez, O.J.; Droscha, C.J.; Singh, S.; Borgman, A.; Lohr, C.E.; Bartlett, P.C.; Taxis, T.M. Identification of BoLA Alleles Associated with BLV Proviral Load in US Beef Cows. Pathogens 2022, 11, doi:10.3390/pathogens11101093.
- Juliarena, M.A.; Barrios, C.N.; Ceriani, M.C.; Esteban, E.N. Hot topic: Bovine leukemia virus (BLV)-infected cows with low proviral load are not a source of infection for BLV-free cattle. Journal of Dairy Science 2016, 99, 4586-4589, doi:10.3168/jds.2015-10480.
- Miyasaka, T.; Takeshima, S.N.; Jimba, M.; Matsumoto, Y.; Kobayashi, N.; Matsuhashi, T.; Sentsui, H.; Aida, Y. Identification of bovine leukocyte antigen class II haplotypes associated with variations in bovine leukemia virus proviral load in Japanese Black cattle. Tissue Antigens 2013, 81, 72-82, doi:10.1111/tan.12041.
- Takeshima, S.N.; Ohno, A.; Aida, Y. Bovine leukemia virus proviral load is more strongly associated with bovine major histocompatibility complex class II DRB3 polymorphism than with DQA1 polymorphism in Holstein cow in Japan. Retrovirology 2019, 16, 10-15, doi:10.1186/s12977-019-0476-z.
- Lo, C.W.; Borjigin, L.; Saito, S.; Fukunaga, K.; Saitou, E.; Okazaki, K.; Mizutani, T.; Wada, S.; Takeshima, S.N.; Aida, Y. BoLA-DRB3 Polymorphism is Associated with Differential Susceptibility to Bovine Leukemia Virus-Induced Lymphoma and Proviral Load. Viruses 2020, 12, doi:10.3390/v12030352.
- Borjigin, L.; Lo, C.-W.; Bai, L.; Hamada, R.; Sato, H.; Yoneyama, S.; Yasui, A.; Yasuda, S.; Yamanaka, R.; Mimura, M.; et al. Risk Assessment of Bovine Major Histocompatibility Complex Class II DRB3 Alleles for Perinatal Transmission of Bovine Leukemia Virus. Pathogens 2021, 10, 502-502, doi:10.3390/pathogens10050502.
- Material and methods, line 92 : 207 compounds were tested, not 227. Besides, it is not possible to check for the compounds tested by the link provided. Please, give any way to have access to the list of compounds tested (for example as a supplementary excell file).
Responses:
Thank you for your very helpful advice. We apologies for the error in the number of compounds. All compound counts have been checked and corrected. Please confirm it in our response of comment 1.
Additionally, according to your suggestion, we added list of second screening as supplemental Table S1 and mention it in the manuscript. Changed previous table S1 to S2 and table S2 to S3 due to this fix. Here are the changes and their locations:
Line 286: “Table 1” to “Tables 1 and S1”
Line 287: “Table S1 and S2” to “Tables S2 and S3”
Line 291: “Table 1” to “Tables 1 and S1”
Line 295: “Table S1 and S2” to “Tables S2 and S3”
Line 300: “Table S1 and S2” to “Tables S2 and S3”
Line 519: “Table S1: Result of second screening;”, added.
Line 520: “Table S2” to “Table S3”
Line 520: “Table S2” to “Table S3”
- Page 3, line 139 : please replace « standard derivative » for « standard deviation »
Response:
Thank you for your gently advice. We apologies for our mistyping and corrected “derivative” to “deviation”.
Line 140: “derivative” to “deviation”
- Page 6 : figure 1 is lacking
Responses:
We apologize for the inconvenience by the loss of figure 1. We added figure 1 to page 6 of the manuscript. Please confirm it. Thank you for your cooperation.
Page 6: figure 1., added
- Table 1 : the sentence about compounds related to BSI-066 at the end of the table has no sense and should be re-written.
Responses:
Thank you for your kindly advise. According to your suggestion, we re-written the sentence at the end of table 1 as below.
Table 1. foot note: “Similar compounds to BSI-066 cannot be obtained the structure-related correlation thereby excluding from the further experiments.” to “We could not find a commonly structure showing sufficiently high activity, so excluded from next experiment.”
- Page 10, line 342, legend to figure 5: please check the letters between parenthesis
Response:
Thank you for your gently advice. We apologies for our mistyping and corrected “B” and “D” to “A” and “C” respectively.
Line 343: “B” to “A”
Line 343: “D” to “C”
- Discussion, line 451 : last sentence of the paragraph shpuld be revised (the expression : « but not BSI-066 » is inadequately located.
Responses:
Thank you for your very helpful advice. According to your suggestion, we deleted “but not BSI-066” from this sentence and added the following sentence after that.
Line 453: “, but not BSI-066”, deleted.
Line 453: “However, we were unable to obtain inhibitors for compounds with structures are similar to BSI-066.”, added.
- Page 13, line 454 : replace « as below » for « as detailed below ».
Responses:
Thank you for your very helpful advice. According to your suggestion, we replace “as below” to “as detailed below”.
Line 459: “as below” to “as detailed below”.
- Page 14, lines 462 to 464 : please revise the construction of this sentence.
Responses:
Thank you for your kindly advise. According to your suggestion, we revised the construction of the sentence as below.
Line 466: “NPDepo contains a large number of flavonoids and isoflavonoids of biochanin A analogs in its library, of which 180 analogs were screened for inhibitory activity and cytotoxicity.” to “NPDepo stocks a large number of flavonoids and isoflavonoids, including biochanin A analogues. In this study, the top 198 analogues with high similarity to biochanin A were used to screen for inhibitory activity.”
- Line 482 : replace « target BSI-625 » for « target of BSI-625 »
Response:
Thank you for your gently advice. We apologies for our mistyping and corrected “target BSI-625” to “target of BSI-625”.
Line 488: “target BSI-625” to “target of BSI-625”
13. Line 496: please check an error of typing.
Response:
Thank you for your gently advice. We apologies for our error of typing and deleted it.
Line 501: “target of f BSI-679” to “target of BSI-679”, deleted
- Reference 23 has errors, please check.
Response:
Thank you for your gently advice. We apologies for error of reference 23. We fixed reference information and checked all references.
Referenece:
“23. Ikebuchi, R.; Konnai, S.; Ohashi, K. Blockade of bovine PD-1 increases T cell function and inhibits bovine leukemia virus expression in B cells in vitro Materials and methods Construction and expression of recombinant soluble bovine PD-1-immunoglobulin fusion protein. 2013, 1-14, doi:10.1186/1297-9716-44-59.” to
“23. Ikebuchi, R.; Konnai, S.; Okagawa, T.; Yokoyama, K.; Nakajima, C.; Suzuki, Y.; Murata, S.; Ohashi, K. Blockade of bovine PD-1 increases T cell function and inhibits bovine leukemia virus expression in B cells in vitro. Vet Res 2013, 44, 59, doi:10.1186/1297-9716-44-59.”
Reviewer 2 Report
Manuscript ID: viruses-2023233
Title: Application of the luminescence syncytium induction assay to identify chemical compounds that inhibit bovine leukemia virus replication
OVERVIEW:
Authors described the application of the luminescence syncytium induction adapted assay (LuSIA-HTS) and water-soluble tetrazolium salt-1 (WST-1) assay to identify the bovine leukemia virus inhibitory activity and toxicity in cell culture of different chemical compounds.
First, they analyzed the inhibition of infectivity and cell viability of 400 compounds and after that selected three of them: BSI-066, BSI-206 and BSI-234.
In a second screening, authors searched for similar compounds to confirm the activity prior determined. They detected that BSI-623 and BSI-625 (similar to BSI-206) and BSI-601, BSI-611 and BSI-613 (similar to BSI-234) had high antiviral activity without toxicity, so they continued their investigation with these compounds.
After that, they looked for commercial compounds and they found BSI-625 and BSI-677 and BSI-601, BSI-613, BSI-678, and BSI-679 as the similar and commercially available to BSI-206 and BSI-234, respectively. Finally, they selected BSI-625 and BSI-679 as the two different structure compounds due to their inhibition activity at low concentrations.
To characterize these two different structural inhibitor candidates, they calculated their inhibitory and cytotoxicity concentration (IC50, CC50) to finally calculated the selective index (SI) of seed compounds as the CC50/IC50.
Authors assayed then, for both compounds, the time-dependent BLV infection inhibitory activity founding that both were found to be most effective when added 0 to 0.5 h after co-cultivation. Consequently, both compunds are involved in early stages of BLV cycle but not in adsorption stages as authors demonstrated by flow cytometry. Authors demonstrated that BSI-625 did not affect viral cDNA reverse transcription comparing to BSI-679 that significant decrease the amount of BLV cDNA indicating that this compound affect in a step before reverse transcription. Finally, after analyzed if both compounds affected BLV infection between viral mRNA transcription and integration, authors found that both compounds have inhibitors function before viral RNA transcription.
In conclusion, authors described here a complete analysis of chemical compounds with antiviral activity using the LuSIA methods proposed by them.
Although, more studies are needed to analyzed the effects of both compounds in vivo, both candidates could be an excellent candidate for therapeutic compounds for BLV infection.
MINOR CORRECTIONS:
Figure 1 is not showed in the text. Please, add it.
Author Response
Response to reviewer 2:
Thank you for your gently review and comments. We appreciate your comments and have amended our manuscript accordingly.
Reviewer 2 comment
Authors described the application of the luminescence syncytium induction adapted assay (LuSIA-HTS) and water-soluble tetrazolium salt-1 (WST-1) assay to identify the bovine leukemia virus inhibitory activity and toxicity in cell culture of different chemical compounds.
First, they analyzed the inhibition of infectivity and cell viability of 400 compounds and after that selected three of them: BSI-066, BSI-206 and BSI-234.
In a second screening, authors searched for similar compounds to confirm the activity prior determined. They detected that BSI-623 and BSI-625 (similar to BSI-206) and BSI-601, BSI-611 and BSI-613 (similar to BSI-234) had high antiviral activity without toxicity, so they continued their investigation with these compounds.
After that, they looked for commercial compounds and they found BSI-625 and BSI-677 and BSI-601, BSI-613, BSI-678, and BSI-679 as the similar and commercially available to BSI-206 and BSI-234, respectively. Finally, they selected BSI-625 and BSI-679 as the two different structure compounds due to their inhibition activity at low concentrations.
To characterize these two different structural inhibitor candidates, they calculated their inhibitory and cytotoxicity concentration (IC50, CC50) to finally calculated the selective index (SI) of seed compounds as the CC50/IC50.
Authors assayed then, for both compounds, the time-dependent BLV infection inhibitory activity founding that both were found to be most effective when added 0 to 0.5 h after co-cultivation. Consequently, both compunds are involved in early stages of BLV cycle but not in adsorption stages as authors demonstrated by flow cytometry. Authors demonstrated that BSI-625 did not affect viral cDNA reverse transcription comparing to BSI-679 that significant decrease the amount of BLV cDNA indicating that this compound affect in a step before reverse transcription. Finally, after analyzed if both compounds affected BLV infection between viral mRNA transcription and integration, authors found that both compounds have inhibitors function before viral RNA transcription.
In conclusion, authors described here a complete analysis of chemical compounds with antiviral activity using the LuSIA methods proposed by them.
Although, more studies are needed to analyzed the effects of both compounds in vivo, both candidates could be an excellent candidate for therapeutic compounds for BLV infection.
Responses:
Thank you for your kindly comment. We agree efficacy of our candidate compounds in vivo is very important. In the future, we would like to clarify the target of these compounds and to clarify their effects in vivo after evaluating their toxicity in mice.
MINOR CORRECTIONS:
Figure 1 is not showed in the text. Please, add it.
Responses:
We apologize for the inconvenience by the loss of figure 1. We added figure 1 to page 6 of the manuscript. Please confirm it. Thank you for your cooperation.
Page 6: Figure 1., added
Reviewer 3 Report
The present study aimed to modify the LuSIA protocol previously report to enable use for high-through put screening, designate LuSIA-HTS, and screen 627 low-molecular-weight compounds from the RIKEN Natural Products Depository.
This study is interesting and I consider that it may be suitable for publication in the journal viruses.
It would have been interesting to have carried out the study of the inhibition of BLV by drugs establishing primary cultures of lymphocytes and the infection of these cultures with an isolate of BLV. Do you consider that the cell lines used in your study are an adequate model for natural BLV infection? Information on this could be included in the discussion.
Author Response
Response to reviewer 3:
Thank you for your gently review and comments. We appreciate your comments and have amended our manuscript accordingly.
Reviewer 3 comment
The present study aimed to modify the LuSIA protocol previously report to enable use for high-through put screening, designate LuSIA-HTS, and screen 627 low-molecular-weight compounds from the RIKEN Natural Products Depository.
This study is interesting and I consider that it may be suitable for publication in the journal viruses.
It would have been interesting to have carried out the study of the inhibition of BLV by drugs establishing primary cultures of lymphocytes and the infection of these cultures with an isolate of BLV. Do you consider that the cell lines used in your study are an adequate model for natural BLV infection? Information on this could be included in the discussion.
Response:
Thank you for your valuable comment.
We also understand the importance of drug efficacy in primary lymphocyte culture cells and virus isolates and consider this to be an issue for the future.
In addition, our previous studies have shown that the cell lines used in this study have a good correlation between viral infection and blood proviral loads in cattle. Therefore, we consider this method to be a good BLV infection model.
Following your kind suggestion, we added the following sentence in discussion section to show our thoughts.
Line 446: “We thought that if we could find a drug that could reduce BLV infection using LuSIA, it may suppress BLV transmission in cattle.”